# Exploring the Mechanism of the Intramolecular Diels–Alder Reaction of (2*E*,4*Z*,6*Z*)-2(allyloxy)cycloocta-2,4,6-trien-1-one Using Bonding Evolution Theory

**DOI:** 10.3390/molecules28196755

**Published:** 2023-09-22

**Authors:** Abel Idrice Adjieufack, Jean Moto Ongagna, Jean Serge Essomba, Monique Bassomo Ewonkem, Mónica Oliva, Vicent Sixte Safont, Juan Andrés

**Affiliations:** 1Laboratory of Theoretical Chemistry (LCT), Namur Institute of Structured Matter (NISM), University of Namur, Rue de Bruxelles, 61, B-5000 Namur, Belgium; 2Physical and Theoretical Chemistry Laboratory, University of Yaoundé 1, Yaoundé P.O. Box 812, Cameroon; jeansergeessomba@gmail.com; 3Computational Chemistry Laboratory, High Teacher Training College, University of Yaoundé 1, Yaoundé P.O. Box 47, Cameroon; 4Department of Chemistry, Faculty of Sciences, University of Douala, Douala P.O. Box 2701, Cameroon; jean.monfils@yahoo.fr (J.M.O.); myewon@gmail.com (M.B.E.); 5Analytical and Physical Chemistry Department, Jaume I University, Avda. Sos Baynat s/n, 12071 Castelló, Spain; oliva@uji.es (M.O.); safont@uji.es (V.S.S.)

**Keywords:** intramolecular Diels–Alder reaction, bonding evolution theory, QTAIM analysis

## Abstract

In the present work, the bond breaking/forming events along the intramolecular Diels–Alder (IMDA) reaction of (2*E*,4*Z*,6*Z*)-2(allyloxy)cycloocta-2,4,6-trien-1-one have been revealed within bonding evolution theory (BET) at the density functional theory level, using the M05-2X functional with the cc-pVTZ basis set. Prior to achieving this task, the energy profiles and stationary points at the potential energy surface (PES) have been characterized. The analysis of the results finds that this rearrangement can proceed along three alternative reaction pathways (a–c). Paths a and b involve two steps, while path c is a one-step process. The first step in path b is kinetically favored, and leads to the formation of an intermediate step, **Int-b**. Further evolution from **Int-b** leads mainly to **3-b1**. However, **2** is the thermodynamically preferred product and is obtained at high temperatures, in agreement with the experimental observations. Regarding the BET analysis along path b, the breaking/forming process is described by four structural stability domains (SSDs) during the first step, which can be summarized as follows: (1) the breaking of the C–O bond with the transfer of its population to the lone pair (V(O)), (2) the reorganization of the electron density with the creation of two V(C) basins, and (3) the formation of a new C–C single bond via the merger of the two previous V(C) basins. Finally, the conversion of **Int-b** (via **TS2-b1**) occurs via the reorganization of the electron density during the first stage (the creation of different pseudoradical centers on the carbon atoms as a result of the depopulation of the C–C double bond involved in the formation of new single bonds), while the last stage corresponds to the non-concerted formation of the two new C–C bonds via the disappearance of the population of the four pseudoradical centers formed in the previous stage. On the other hand, along path a, the first step displays three SSDs, associated with the depopulation of the V(C2,C3) and V(C6,C7) basins, the appearance of the new monosynaptic basins V(C2) and V(C7), and finally the merging of these new monosynaptic basins through the creation of the C2–C7 single bond. The second step is described by a series of five SSDs, that account for the reorganization of the electron density within **Int-a** via the creation of four pseudoradical centers on the C12, C13, C3 and C6 carbon atoms. The last two SSDs deal with the formation of two C-C bonds via the merging of the monosynaptic basins formed in the previous domains.

## 1. Introduction

Intramolecular cyclic rearrangements refer to the reaction of a single molecule where two atoms or sites react to form a new cyclic product. Among them, the intramolecular Diels–Alder (IMDA) reaction is widely used for the stereoselective synthesis of complex molecules containing fused and/or bridged 6-membered rings, which appear in many natural products or pharmaceuticals [1,2,3]. IMDA reactions are faster, cleaner and more selective than intermolecular reactions [4]. In the seminal review on IMDA reactions, Brieger and Bennett [3], based on the results by Kitahara et al. [5], reported on the IMDA reaction of (2*E*,4*Z*,6*Z*)-2(allyloxy)cycloocta-2,4,6-trien-1-one, **1** to **2** (path a), although other products, **3b-1** and **3b-2**, and **4c-1** and **4c-2**, can be also formed via path b and path c, respectively (see Figure 1).

Path **a** involves two steps: in the first one, cyclooctatriene **1** yields the bicyclic species 6-(allyloxy)bicyclo [4.2.0]octa-2,4-dien-7-one, which in turn rearranges to **2** via an IMDA process. Path **b** also proceeds along a two steps mechanism: in the first one, **1** undergoes a [3,3] sigmatropic rearrangement leading to (4*Z*,6*Z*)-3-allylcycloocta-4,6-diene-1,2-dione, which in turn suffers an IMDA process leading either to **3-b1** or to **3-b2**. Path **c** is just the result of the direct Diels–Alder reaction of **1** to either **4-c1** or **4-c2**.

One of the ultimate goals of chemistry is to understand how chemical bonds break/form throughout the progress of a chemical reaction, which in turn implies the ability to disclose the underlying mechanism at an atomic scale. In his seminal works on quantum theory of atoms in molecules (QTAIM) [6,7,8], Richard Bader has demonstrated that topological analysis of the electron density, ρ(r), as a quantum chemically accessible scalar field, condenses the chemically relevant information obtained from quantum calculations into an observable computed from it, such as electron density. Later, Popelier introduced the concept of quantum chemical topology (QCT) [9,10] to embrace QTAIM, bonding evolution theory (BET) and non-covalent interaction analysis (NCI), as appropriate tools to analyze the topology of the electron density ρ(r), by means of real-space partitioning of the molecular space by using functions of the electronic density and/or its derivatives [11,12,13]. Within the BET framework, the evolution of the topology of the ELF along a chosen reaction path (e.g., the intrinsic reaction coordinate, connecting the reactants to the products) is characterized in terms of Thom’s elementary catastrophes. BET has a demonstrated capability that not only distinguishes between fundamental changes to ρ(r) electron density throughout a chemical reaction, but also stablishes where and how the chemical bonds are broken throughout the reaction progress [11,14,15,16,17,18,19,20,21,22,23]. These processes become naturally associated with specific stability domains (SSDs) separated by catastrophe bifurcations [14,16,24,25,26,27].

In view of the scope of the IMDA reaction, computational studies on its mechanism are important in the areas of theoretical and synthetic organic chemistry. However, few computational/theoretical studies applying BET to the corresponding mechanisms have been published to date [28,29]. Herein, we report on a theoretical study, based on BET, to disclose the nature of the reaction mechanisms for the three possible reactive channels for the transformation of **1** (Figure 1). Specifically, the answers to the following questions are the main goals of the present work: (1) where and how do electron density changes occur during the reaction, (2) how can electron density rearrangement track events in the bond breaking/forming process, and (3) how should the electronic reorganization along the chemical reaction path be deciphered? Or, in other words, what types of catastrophes and SSDs appear throughout each reaction pathway during BET analysis?

## 2. Results and Discussion

### 2.1. Thermodynamic and Geometrical Aspects

The IMDA reaction of (2*E*,4*Z*,6*Z*)-2(allyloxy)cycloocta-2,4,6-trien-1-one (**1**) can proceed, as explained, along three reaction paths (a, b and c) and leads to the formation of 6-(allyloxy)bicycle [4.2.0]octa-2,4-dien-7-one and (4*Z*,6*Z*)-3-allylcyclooctane-4,6-dien-1,2-dione, named as intermediates **Int-a** and **Int-b**, respectively, together with the final products, namely **2**, **3-b1**, **3-b2**, **4-c1** and **4-c2** (see Figure 2).

The reaction mechanism along path a is divided into two steps, the first leads to **Int-a**, while the second step corresponds to the Diels–Alder process. The first step corresponding to the tautomerization of (2*E*,4*Z*,6*Z*)-2(allyloxy)cycloocta-2,4,6-trien-1-one and leading to the formation of **Int-a** overcomes an activation free Gibbs energy of 56.6 kcal/mol with a reaction Gibbs free energy of 13.3 kcal/mol (see Figure 1 and Table 1). The transformation of **Int-a** to **2** presents an activation Gibbs free energy of 12.0 kcal/mol and a reaction Gibbs free energy of 40.1 kcal/mol. Path b starts with a [3,3] sigmatropic rearrangement of **1** leading to **Int-b**, followed by its conversion into **3-b1** and **3-b2**. This [3,3] sigmatropic rearrangement needs to overcome an activation barrier of 31.3 kcal/mol and has a reaction energy of −5.6 kcal/mol. During the second step, **Int-b** can perform two alternative Diels–Alder processes through the transition state **TS2-b1** (with a barrier of 46.1 kcal/mol) or **TS2-b2** (with a barrier of 64.0 kcal/mol) to form two cycloadducts, **3-b1** and **3-b2**, having a reaction Gibbs free energy of −10.1 and 31.2 kcal/mol, respectively, by considering **Int-b** as a reference. Thus, **3-b1** is more thermodynamically stable than **3-b2**, since its formation energy is 41.3 kcal/mol lower (see Figure 1). Finally, the last path, path c corresponds to a direct Diels–Alder reaction of (2*E*,4*Z*,6*Z*)-2(allyloxy)cycloocta-2,4,6-trien-1-one via the activation Gibbs free barrier of 63.9 (**TS1-c1**) and 70.0 kcal/mol (**TS1-c2**) leading to the formation of cycloadducts, **4-c1** and **4-c2**, that takes place with a reaction energy of 0.7 and 20.6 kcal/mol, respectively. As can be seen, our results confirm **2** as the thermodynamically preferred product at high temperatures, as experimentally detected [5], while **3-b1** is the main product in kinetically controlled conditions.

Figure 2 displays the geometrical structures of the different transition states (TSs) during each reaction pathway. The key distances of the new forming bonds are indicated. Along the reaction path a, they are equal to 2.143 Å at **TS1-a**, and 2.240 and 2.229 Å at **TS2-a**. Concerning path b, they correspond to 1.942 (C–O) and 2.116 (C–C) Å at **TS1-b**, 2.180 (C–C) and 2.170 (C–C) Å at **TS2-b1**, and 1.938 (C–O) and 1.875 (C–O) Å at **TS2-b2**. Finally, the length of the two new forming C–C bonds are equal to 2.271 and 2.309 Å at **TS1-c1**, and 2.345 and 1.992 Å at **TS1-c2**.

### 2.2. QTAIM Analysis of the Transition State Structures

Before performing the analysis of bond forming/breaking processes, an AIM study on each transition state is required. Figure 3 displays the presence of bond critical points (BCPs), while Appendix A contains the value of each topological parameter. According to Appendix A, each TS structure presents a small value (ca. 0.12 a.u. or less) on the Laplacian density at the different BCPs, corresponding to a type of non-covalent or closed-shell (ionic) interactions. Moreover, the positive value of the Laplacian density confirms the initial stage in the formation of various new single bonds (see Section 2.3).

In addition to the topological parameters discussed above, the other BCP parameters were used to characterize the chemical bond properties of the TS structures for the IMDA reaction. However, with respect to the total energy density H(r), all the values are negative (Appendix A), which at first glance might suggest significant electron sharing. The highest magnitude values obtained (−0.02 a.u. for $O_{10}\ldots C_{11}$ at **TS1-b**, −0.032 and −0.022 a.u. for $O_{10}\ldots C_{12}$ and $O_{9}\ldots C_{11}$ at **TS2-b2**, respectively) reflect a higher degree of covalence in these interactions. Likewise, in order to measure the π character of the bond, ellipticity descriptors have been admitted. Thus, the $C_{4}\ldots C_{12}$ bond of **TS2-b1** exhibits a higher ellipticity (0.638 a.u.) than all the others, suggesting that it is more involved in a hyper conjugative interaction than in the case of the other TSs and corresponds to a greater instability of this bond [30]. Furthermore, since some covalence is evidenced by the ratio |*V*(r)|/*G*(r), which is always greater than 1 at the BCP, the emerging C–C and C–O bonds seem to be triggered more easily in all the TS series (ratio close to 1.5) (Appendix A). Thereafter, the QTAIM analysis shows that along the TS series, passing from path a to path c, the covalent character of all the bonds are reinforced, since the values of the ratio |λ1|/λ3 increase (values close to 0.35) (Appendix A).

### 2.3. BET Analysis along Different Reaction Paths

As we have shown in Section 2.1, the IMDA reaction of (2*E*,4*Z*,6*Z*)-2(allyloxy)cycloocta-2,4,6-trien-1-one occurs along three reaction paths. Therefore, a BET study is carried out to gain deep insight into the corresponding bond breaking/forming processes.

#### 2.3.1. BET Analysis within Path a

##### First Step: Tautomerization Process Yielding **Int1-a**

Analysis of the results presented in Figure 4 and Appendix A reveals that three structural stability domains (SSDs) are required to describe the formation of the C2–C7 bond. The first domain, SSD-I (d(C2–C7) = 2.606 Å) displays the electron population of the key atoms involved in the bond formation: the disynaptic V(C2,C3), V(C4,C5) and V(C6,C7) basins, which illustrate the C2–C3, C4–C5 and C6–C7 double bonds, and hold an electron population of 3.63, 3.42 and 3.48e, respectively, at the beginning of the domain. In addition, two other disynaptic basins V(C3,C4) and V(C5,C6) with a population of 2.12e at the beginning, symbolize the single C3–C4 and C5–C6 bonds, respectively. At the end of this domain, some electron fluctuations are recorded with a decrease in the population of 0.67, 0.57 and 0.60e for the V(C2,C3), V(C4,C5) and V(C6,C7) basins, respectively. In fact, these drops in basin population are mainly transferred to the V(C3,C4) (+0.87e) and V(C5,C6) (+0.63e) basins.

At the beginning of the second domain, SSD-II (d(C2–C7) = 2.067 Å), the electron population of the V(C3,C4) and V(C5,C6) basins continue to increase, while the population of the V(C2,C3), V(C4,C5) and V(C6,C7) basins decrease. This continuous decrease in the populations of the V(C2,C3) and V(C6,C7) basins comes from the appearance of two new monosynaptic basins (V(C2) and V(C7), see Figure 5) on the C2 and C7 carbon atoms, with an electron population of 0.31 ad 0.26, respectively. These monosynaptic basins are the precursors [31,32] for the formation of a new C–C single bond.

The new single C–C bond appears at the beginning of the last domain, SSD-III (d(C2–C7) = 1.859 Å), from the merger of two former V(C2) and V(C7) basins formed at the SSD-II domain. The electron population starts from 1.44e, before reaching a value of 1.92e at the end of the domain (d(C2–C7) = 1.569 Å).

Extending from −15.72 to 6.74 amu^1/2^ Bohr, the topological changes occur along the IRC path at the following reaction coordinates: −0.48 and −1.76 amu^1/2^ Bohr, allowing for the calculation of the value of synchronicity ($S_{y}$) and absolute synchronicity ($S_{y}^{abs}$), which are equal to 0.94 and 0.91, respectively. According to these values, the topological changes take place with 91% of synchronous character [33,34].

##### Second Step: Diels–Alder Reaction of the Intermediate **Int-a** Yielding to **2**

The BET analysis of the Diels–Alder reaction of the intermediate **Int-a** yielding to **2** is described by five SSDs (see Figure 6 and Appendix A). The first domain, SSD-I (d(C3–C12) = 3.235 Å and d(C6–C13) = 3.655 Å), represents the electron population of different atoms of the intermediate **Int-a**, required for the formation of the new C3–C12 and C6–C13 single bonds (see Appendix A). The transition from SSD-I to II (d(C3–C12) = 2.208 Å and d(C6–C13) = 2.184 Å) deals with the creation of two-fold catastrophes on the C12 and C13 carbon atoms. In fact, these two-fold catastrophes correspond to the creation of monosynaptic basins, whose electron population of 0.21 and 0.25e come from the reduction in the V(C12,C13) basin population. The electron populations of the V(C12) and V(C13) basins slightly increase at the beginning of the third domain, SSD-III (d(C3–C12) = 2.166 Å and d(C6–C13) = 2.137 Å), while we note a high decrease in the electron population of the V(C3,C4) and V(C5,C6) basins. These electron drops illustrate the appearance of another two new monosynaptic basins (V(C3) and V(C6)) integrating an electron population of 0.29 and 0.26 e, respectively.

At the beginning of SSD-IV (d(C3–C12) = 1.997 Å and d(C6–C13) = 1.954 Å), the population of the V(C3) and V(C12) basins continues to grow, while that of the V(C6) and V(C13) basins has completely disappeared. The new V(C6,C13) basin collects its population of 1.20e from the former population of V(C6) and V(C13) (0.51 and 0.56e, respectively, at the end of the last domain). Finally, at the beginning of the last domain SSD-V (d(C3–C12) = 1.911 Å and d(C6–C13) = 1.863 Å), the formation of the second C3–C12 single bond, associated with the appearance of a cusp catastrophe, begins when the formation of the former C3–C4 single bond has reached 77% of its population. This last cusp catastrophe corresponds to the creation of the V(C3–C12) disynaptic basin with a population of 1.28e, which symbolizes the formation of the new C3–C12 single bond.

With the values of the reaction coordinates (0.30, 0.60, 1.79 and 2.40 amu^1/2^ Bohr) for the different topological changes along this pathway, we have calculated the values of $S_{y}$(0.93) and $S_{y}^{abs}$(0.88). Therefore, these bond formation processes take place with 88% of synchronous character.

#### 2.3.2. BET Analysis within Path b

##### First Step: [3,3] Sigmatropic Rearrangement of **1** Yielding to **Int-b**

The [3,3] sigmatropic rearrangement of **1** via transition state **TS1-b** yields the intermediate **Int-b**, which is described by a series of four SSDs (see Figure 7 and Appendix A). The electron population of the key basins engaged in the formation of a new C3–C13 bond are given in Appendix A. Accordingly, the first domain SSD-I (d(O10–C11) = 1.444 Å and d(C3–C13) = 3.826 Å) shows the presence of five disynaptic (V(C2,C3), V(C2,O10), V(O10,C11), V(C11,C12) and V(C12,C13)) and one monosynaptic (V(O10)) basins related with the process, while the V(O10,C11) basin is not present at the beginning of the second domain, SSD-II (*d*(O10–C11) = 1.696 Å and d(C3–C13) = 2.313 Å). The non-presence of this basin illustrates the rupture of the O10–C11 bond with its electron population transferred to the O10 lone pair. In fact, the electron population of the V(O10) basin suddenly increases by +0.59e and almost equals the former population of 0.63e of the V(O10,C11) basin that disappeared at the beginning of the second domain (SSD-II, d(O10–C11) = 1.696 Å and d(C3–C13) = 2.313 Å).

Compared to the previous domain, the population of the V(O10) basin slightly decreases by 0.43e at the beginning of the third domain (SSD-III, d(O10–C11) = 1.989 Å and d(C3–C13) = 2.075 Å), while at the same time the V(C2,O10) gains 0.34e. This increase in the population of V(C2,O10) reflects the transformation of the C2-O10 single bond into a double one. However, we also note the presence of two new fold-catastrophes via the appearance of the V(C3) and V(C13) basins. They are populated by 0.32 and 0.23e, respectively, and come from the reduction in the population of the V(C2,C3) and V(C12,C13) basins. These new populations reach up to 0.52 and 0.36e at the end of the domain, before their merger at the beginning of the last domain (SSD-IV, d(O10–C11) = 2.162 Å and d(C3–C13) = 1.889 Å). In fact, this merger allows the creation of the new disynaptic basin V(C3,C13) with a population almost equal to those of the two former V(C3) and V(C13) basins. It starts with a population of 1.03e and reaches 1.84e at the end of the domain. Analysis of the results in Appendix A and Appendix A reveals that the increase in the V(C11,C12) and V(C2,O10) populations corresponds to the transformation from single to double bonds, while and inverse process is obtained by the decrease in the V(C2,C3) and V(C12,C13) populations.

Along this IRC path (varying from −14.15 to 15.41 amu^1/2^ Bohr), the different topological changes take place at the following reaction coordinates: −1.57, 0.31 and 1.26 amu^1/2^ Bohr. The corresponding values of $S_{y}$ and $S_{y}^{abs}$ are 0.94 and 0.90, respectively. The last parameter implies that the topological changes took place along the path with 90% of synchronous character.

##### Second Step: Diels—Alder Reaction of Intermediate **Int-b** Yielding **3-b1** and **3-b2**

The conversion of intermediate **Int-b** into **3-b1** and **3-b2** via the transition states **TS2-b1** and **TS2-b2** was also analyzed and a series of six and three SSDs were required to describe the bond breaking and forming processes during this chemical transformation, see Appendix A and Appendix A. As for **TS2-b1**, along SSD-I, the population of V(C5,C6) increases, while the populations of V(C4,C5), V(C6,C7) and V(C11,C12) decrease. The next three domains SSD-II (d(C4–C12) = 2.170 Å and d(C7–C11) = 2.180 Å), SSD-III (d(C4–C12) = 2.137 Å and d(C7–C11) = 2.144 Å) and SSD-IV (d(C4–C12) = 2.103 Å and d(C7–C11) = 2.108 Å) further describe the processes of the C4–C5, C6–C7 and C11–C12 double bonds into singles and the creation of the pseudoradical centers on the C7, C11, C12 and C4 carbon atoms. These pseudoradical centers are illustrated by the appearance of the V(C7), V(C11), V(C12) and V(C4) monosynaptic basins, whose populations come directly from the reduction in the population of the disynaptic basins, V(C6,C7), V(C11,C12) and V(C4,C5). The beginning of the fifth domain SSD-V (d(C4–C12) = 1.968 Å and d(C7–C11) = 1.967 Å), starts with the appearance of the V(C7,C11) basin due to the merger of V(C7) and V(C11) highly populated in domain IV. At the same time, the populations of the V(C4) and V(C12) basins slightly increase in order to prepare for the appearance of the main last cusp catastrophe (V(C4,C12)). It appears at the beginning of SSD-VI, as the materialization of the formation of the last C4–C12 bond (see Figure 8 and Appendix A, as well as Appendix A).

According to BET analysis, the second and third domains along the **TS2-b2** path involve the presence of the two disynaptic V(O10,C12) and V(O9,C11) basins, which illustrate the formation of new O10–C12 and O9–C11 single bonds. In fact, at the end of the first domain, the population of the V(O9) and V(O10) basins record a slight increase of 0.51 and 0.61e, respectively. At the same time, the population of the V(C1,O9) and V(C2,O10) basins are strongly depopulated (0.72 and 0.88e), as well as V(C11,C12), which loses 1.22e, in favor of the V(C1,C2) basin, which has recorded an increase of 1.60e, as well as the monosynaptic V(O9) and V(O10) basins (whose populations increase by 0.51 and 0.61e, as already mentioned). The next two final steps describe the formation of the O10–C12 (SSD-II) and O9–C11 (SSD-III) bonds with the population of 0.64 and 0.79e, coming from the reduction of the main lone pairs on O10 and O9, which have lost a population of 0.54 and 0.75e, respectively (see Figure 9 and Appendix A, as well as Appendix A). Their populations reach up to 1.21 or 1.22e at the end of the last domain, while the population of the V(C1,C2) basin is worth 4.00e, and symbolize the total transformation of the C1–C2 single bond into a double bound.

The corresponding values of $S_{y}^{abs}$ along the two paths are equal to 0.95 and 0.97, and these latter values predict that the topological changes along the **TS2-b2** pathway are slightly more synchronous compared to the changes in the **TS2-b1** pathway.

#### 2.3.3. BET Analysis within Path c

The Diels–Alder reaction of **1** yielding to products **4-c1** and **4-c2** via the transition states **TS1-c1** and **TS1-c2** takes place within five SSDs, as displayed in Figure 10 and Figure 11, while the electron populations of the key basins engaged in the formation of two new C–C bonds are given in Appendix A and their evolution throughout the processes are illustrated in Appendix A. Along the **TS1-c1** reaction path, the second domain (d(C4–C12) = 2.274 Å and d(C7–C13) = 2.233 Å) deals with the depopulation of the main V(C12,C13) basin via the creation of the V(C13) monosynaptic basin, which integrates a population of 0.23e, while the third domain (d(C4–C12) = 2.239 Å and d(C7–C13) = 2.196 Å) depicts the depopulation of the V(C4,C5) basin due to the appearance of the new V(C4) basin. Like the V(C13) basin, the V(C4) basin with a population of 0.30e, represents the pseudoradical center on the C4 atom, and is required for the formation of the new single C4–C12 bond. In addition to the new V(C4) basin, we also note the appearance of another two new monosynaptic basins, namely V(C7) and V(C12), with a population of 0.33 and 0.29e at the beginning of the domain before reaching 0.77 and 0.60e, respectively, at the end of the domain, at the expense of the V(C6,C7) and V(C12,C13) basins. At the same time, the population of the V(C4) and V(C13) basins reaches 0.63 and 0.55e.

The high population density of different monosynaptic basins starts to disappear along the last two domains (SSD-IV and V). The beginning of the fourth domain (SSD-IV, d(C4–C12) = 1.989 Å and d(C7–C13) = 1.933 Å) starts with the disappearance of the V(C7) and V(C13) basins due to their merger into a new synaptic basin, V(C7,C13). This new V(C7,C13) basin collects its population of 1.41e from the former populations of the V(C7) and V(C13) basins. Finally, the second disappearance deals with the appearance of last cusp catastrophe SSD-V (d(C4–C12) = 1.953Å and d(C7–C13) = 1.896 Å) corresponding to the presence of the V(C4,C12) basin with a population of 1.38e. Moreover, this presence illustrates the formation of a new single C4–C12 bond, see Figure 10 and Appendix A.

For the **TS1-c2** pathway, the second domain SSD-II (d(C2–C12) = 1.992 Å and d(C5–C13) = 2.345 Å) starts with the presence of two new monosynaptic basins, V(C2) and V(C12). The V(C13) basin appears at the beginning of the next domain SSD-III (d(C2–C12) = 1.957 Å and d(C5–C13) = 2.313 Å), as well as the V(C5) basin. The presence of these four monosynaptic basins illustrates the formation of the two new C–C single bonds. They appear at the beginning of the SSD-IV (d(C2–C12) = 1.854 Å and d(C5–C13) = 2.217 Å) and SSD-V (d(C2–C12) = 1.693 Å and d(C5–C13) = 2.044 Å) domains, with the C2–C12 bond followed by the C5–C13 bond.

Along the first pathway (**TS1-c1**), the different changes take place at 0.25, 0.51, 2.29 and 2.54 amu^1/2^ Bohr reaction coordinates, while the corresponding values of $S_{y}$ and $S_{y}^{abs}$ are equal to 0.94 and 0.89. Like the previous reaction path, the **TS1-c2** pathway presents a synchronous character (93%), which is 4% higher than the corresponding value along the **TS1-c1** pathway.

## 3. Computational Method

The geometries of the reactant, intermediates, transition states and products involved in the IMDA reaction were optimized with the M05-2X DFT functional [35] coupled with the cc-pVTZ basis set, as implemented in the Gaussian 16 program [36]. A previous benchmarking study was carried out for the TSs of path a corresponding to the formation of product **2** in the gas phase and at 195 °C. We have assessed that M05-2X performs, as a whole, better than the other tested methods.

The frequency calculations were performed with the experimental conditions [5] of T = 468.15 K and P = 1 atm. The solvent effects (diphenyl ether) were included through the PCM method [37] on the gas-phase optimized geometries. All the transition state structures were characterized by only one imaginary frequency, while the minima (reactant, intermediates and products) showed a real frequency mode. The intrinsic reaction coordinate (IRC) [38] curves were calculated using the second-order Gonzalez–Schlegel integration algorithm [39,40] to confirm the energy profile connecting each TS to the two minima in the proposed reaction mechanism.

To assess the bonding properties of the forming C–C/C–O single bonds taking place throughout this intramolecular Diels–Alder reaction, AIM analysis within the QTAIM framework was performed by using the Multiwfn program [41] at the M05-2X/cc-pVTZ level. Finally, to evaluate the formation process for these new bonds, the ELF topological analysis was carried out along the IRC curve by extracting the corresponding wave function at each point of the IRC. The ELF was calculated through the TopMod package [42] with a grid step of 0.2 Bohr, while the ELF basin positions along the IRC were visualized with DrawProfile 1.5.5(2471) [43].

## 4. Conclusions

The IMDA reaction of (2*E*,4*Z*,6*Z*)-2(allyloxy)cycloocta-2,4,6-trien-1-one has been studied by means of BET, using the M05-2X/cc-pVTZ computation level. The breaking/forming processes along the complete PES is analyzed in detail. This IMDA rearrangement takes place along three alternative paths (a–c) to yield the adducts, **2**, **3**-**b**(**1**-**2**) and **4-c**(**1**-**2**). Within the kinetically favorable channel (path b), the first step along the **TS1-b** pathway presents a Gibbs free activation energy of 31.3 kcal/mol, which is 25.3 kcal/mol lower than that of **TS1-a** (yielding finally to product **2**), and also 32.6 and 38.7 kcal/mol lower compared to **TS1-c1** and **TS1-c2**, respectively. From **TS1-b**, **Int-b** would be formed and finally through **TS2-b1** the product **3-b1** would be found at −15.7 kcal/mol. However, under thermodynamic control conditions, at high temperatures, the preferred product is predicted to be **2**, that lies 26.8 kcal/mol under the reactant energy, in agreement with the experimental observations.

Concerning the properties of each new forming bond in the TS structure, the AIM analysis reveals a positive value of the density and its Laplacian, as a consequence of the non-formation of the new bonds (C–C and C–O) within different transition states.

Firstly, for the bond forming process along reaction path a, the first step (**TS1-a**) displays three SSDs, listed as follows: the depopulation of the V(C2,C3) and V(C6,C7) basins via the appearance of the new monosynaptic basins V(C2) and V(C7), and finally the merger of these two new monosynaptic basins through the creation of a new synaptic V(C2,C7) basin, associated with the formation of a new C–C single bond. The next step (**TS2-a**), dealing with the conversion of **Int-a** into **2,** is described by a series of five SSDs. SSDs II and III describe the reorganization of the electron density within **Int-a** via the creation of four pseudoradical centers on the C12, C13, C3 and C6 carbon atoms through the appearance of the V(C12), V(C13), V(C3) and V(C6) basins, respectively. Moreover, the last two SSDs deal with the formation the C–C bond via the presence of the different monosynaptic basins formed in the two previous domains (II and III).

Secondly, along reaction path b, a series of four SSDs is required to describe the bond breaking and forming process during the first step (**TS1-b**), while a series of six and three SSDs are required for the formation process of the C–C and C–O single bonds upon the conversion of **Int-b** into **3-b1** and **3-b2**, respectively. Along the path involving **TS1-b**, the first step begins with the breaking of the C–O bond and the transfer of its population to the lone pair O10, while the second step illustrates the reorganization of the electronic density with the creation of the V(C3) and V(C13) basins, required for the last catastrophe, V(C3,C13) basin, corresponding to the formation of the C–C bond. The second step along the **TS2-b1** pathway comprises the creation of pseudoradical centers on the various carbon atoms engaged in the formation of new C–C bonds. They proceed in the last domains with the formation of the C7–C11 bond, followed by the C4–C12 bond. For the **TS2-b2** pathway, the SSD-II begins with the formation of the O10-C12 bond due to the reorganization of the electronic density around the two O9 and O10 lone pairs and the C1–C2 bond due to the appearance of the V(O10,C12) basin, illustrating the new O10-C12 single bond, and ends up with the formation of the O9-C11 single bond.

Finally, the BET analysis of path c revealed that five SSDs are required to describe the formation of the two C–C bonds in **TS1-c1** and **TS1-c2**. The first two stages involve the creation of a pseudoradical center on the carbon atoms, while the last two correspond to the formation of two new single bonds (C–C).

## Data Availability

No new data were created, apart from that reported as Appendix A.

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
