# Peer review of "Exploring the Mechanism of the Intramolecular Diels–Alder Reaction of (2E,4Z,6Z)-2(allyloxy)cycloocta-2,4,6-trien-1-one Using Bonding Evolution Theory"

_molecules, 2023, doi:10.3390/molecules28196755_

Round 1

Reviewer 1 Report

molecules-2587280 is an interesting investigation performed by experts in the field. Yet two comments should be addressed:

1) why using the M05-2X XCF? Is it particularly good for the present IMDA? Do the authors have assess its performance?

2) the calculations are reported for a T = 468K and P = 1 atm, to match the experimental conditions. First, could the authors confirm that this is a gas phase reaction? Secondly, what are the experimental data to compare with and how the calculations could help explaining the experimental results? 

Reviewer 2 Report

The manuscript presented by Juan Andrés and coworkers contains enough information for publication in molecules. It describes theoretical studies on the bond breaking/forming events along the intramolecular Diels-15 Alder (IMDA) reaction of (2E,4Z,6Z)-2(allyloxy)cycloocta-2,4,6-trien-1-one have been revealed 16 within Bonding Evolution Theory (BET) at the density functional theory level, M05-2X functional with cc-pVTZ basis set. The work could be accepted for publication in its current form after adding a suitable reason for the use of the M05-2X functional with cc-pVTZ basis sets for the studied systems. In addition, the style of references should be modified to MDPI references style.

The quality of English is good and readable

Reviewer 3 Report

The manuscript entitled "Exploring the Mechanism of the Intramolecular Diels-Alder Reaction of (2E,4Z,6Z)-2(allyloxy)cycloocta-2,4,6-trien-1-one Using Bonding Evolution Theory" is well written and effectively presented, making it easily understandable.

However, I would like to suggest a few areas that the author may consider addressing:

Firstly, in the concluding remarks, it would be beneficial to provide a stronger emphasis on the thermodynamic aspects of the reactions. This could enhance the overall understanding of the implications of the study and its potential applications.

Furthermore, it would be insightful if the authors could comment on the potential dependency of the reaction paths on the choice of the employed density functional theory (DFT) functional. Explaining the reasoning for choosing M05-2X over other DFT functionals would give additional validation to the research findings.

Please note that these suggestions are intended to enhance the manuscript and strengthen its scientific value. Overall, the manuscript showcases well-presented research that contributes to the field of intramolecular Diels-Alder reactions.
